# Optimizing Geometry and ETL Materials for High-Performance Inverted Perovskite Solar Cells by TCAD Simulation

**DOI:** 10.3390/nano14151301

**Published:** 2024-08-02

**Authors:** Irodakhon Gulomova, Oussama Accouche, Rayimjon Aliev, Zaher Al Barakeh, Valikhon Abduazimov

**Affiliations:** 1Renewable Energy Sources Laboratory, Andijan State University, Andijan 170100, Uzbekistan; irodakhon.gulomova@yahoo.com (I.G.);; 2College of Engineering and Technology, American University of the Middle East, Egaila 54200, Kuwait

**Keywords:** perovskite, inverted structure, thickness, metal oxides, photoelectric parameters

## Abstract

Due to the optical properties of the electron transport layer (ETL) and hole transport layer (HTL), inverted perovskite solar cells can perform better than traditional perovskite solar cells. It is essential to compare both types to understand their efficiencies. In this article, we studied inverted perovskite solar cells with NiO_x_/CH_3_NH_3_Pb_3_/ETL (ETL = MoO_3_, TiO_2_, ZnO) structures. Our results showed that the optimal thickness of NiO_x_ is 80 nm for all structures. The optimal perovskite thickness is 600 nm for solar cells with ZnO and MoO_3_, and 800 nm for those with TiO_2_. For the ETLs, the best thicknesses are 100 nm for ZnO, 80 nm for MoO_3_, and 60 nm for TiO_2_. We found that the efficiencies of inverted perovskite solar cells with ZnO, MoO_3_, and TiO_2_ as ETLs, and with optimal layer thicknesses, are 30.16%, 18.69%, and 35.21%, respectively. These efficiencies are 1.5%, 5.7%, and 1.5% higher than those of traditional perovskite solar cells. Our study highlights the potential of optimizing layer thicknesses in inverted perovskite solar cells to achieve higher efficiencies than traditional structures.

## 1. Introduction

Today, the demand for energy is increasing rapidly. To meet the growing need for electrical energy while preserving the environment and resources, we must turn to renewable energy sources. Among these, wind and solar energy are particularly popular. Solar cells are used to convert solar energy into electrical energy. Currently, there are four generations of solar cells, each aiming to address the limitations of the previous generation. Solar cells have three main types of losses: optical [1], electrical [2], and thermal [3]. Optical losses include reflection, spectral mismatch [4], parasitic absorption [5], and transmission. To reduce reflection, textures [6] and antireflection coatings [7] are applied. Depending on the refractive index of the antireflection coating, the reflection coefficient can be reduced by more than 20%. Spectral mismatch occurs because solar cells primarily absorb photons with energy higher than the material’s band gap. High-energy photons generate hot carriers, which are unstable and recombine quickly through a process called thermalization, leading to parasitic absorption and heating of the solar cell. Due to spectral mismatch and parasitic absorption, the efficiency of single-junction solar cells cannot exceed 33%, according to the Shockley–Queisser limit [8]. Tandem [9] and heterojunction structures [10] offer solutions to spectral mismatch, with theoretical efficiencies of up to 68% achievable using tandem cells with an infinite number of stacked cells [11]. Another way to improve the efficiency of solar cells is inclusion of metal nanoparticles to use from the nanoplasmonic effect [12].

In industry, silicon solar cells are predominantly produced. However, in the last 15 years, there has been growing interest in perovskite solar cells. Perovskite materials possess unique properties that make them excellent candidates for photovoltaic devices, such as tunable band gaps by mixing cations and anions [13], shallow traps [14], long diffusion lengths [15], and high absorption [16] in thin layers. Despite their potential, perovskite solar cells are not yet widely manufactured due to their low stability [17] and size effect [18]. The main causes of degradation in perovskite solar cells are structural distortion [19] under temperature and illumination, as well as ion migration [20]. Various methods have been proposed to improve their stability. Traditional perovskite solar cells have a design where light hits the electron transport layer first, while inverted perovskite solar cells have a reversed design, with light hitting the hole transport layer first. The main reason for creating inverted solar cells is to improve their stability and make them easier to produce. This design helps solve problems like degradation and sensitivity to moisture and oxygen, making the cells last longer. Inverted perovskite solar cells have advantages like better stability, simpler production, and compatibility with flexible substrates. Current research of inverted perovskite solar cells is focused on improving their efficiency and long-term durability to make them even more competitive with other types of solar cells.

The architecture of a perovskite solar cell consists of an electron transport layer (ETL), light-absorbing layers (perovskite), and a hole transport layer (HTL). In our previous work [21], we studied a ZnO/perovskite/NiO_x_ structure and found that the optical properties of NiO_x_ might be superior to ZnO, serving both as an antireflection coating and a charge transport layer. In this study, we decided to investigate an inverted perovskite solar cell with an HTL/perovskite/ETL structure. We used NiO_x_ as the HTL and ZnO, MoO_3_, and TiO_2_ as the ETL. According to the band structure of perovskite solar cells (as shown in Figure 1), ZnO, MoO_3_, and TiO_2_ are excellent choices for the ETL because they do not block electrons. In thin layers, the thickness of each layer significantly affects the performance of the solar cell [22]. Therefore, we focused on optimizing each layer of the inverted perovskite solar cell and identifying the optimal material for the ETL. Additionally, we compared inverted and traditional perovskite solar cells to determine which configuration performs better.

## 2. Materials and Methods

There are three primary methods employed in solar cell research: theoretical analysis, experimental investigation, and numerical simulation. Over the years, simulation techniques have seen significant advancements and gained popularity. Among the various simulation tools used for solar cell research—Silvaco TCAD, Sentaurus TCAD, SCAPS-1D, and Comsol Multiphysics—Sentaurus TCAD stands out due to its extensive range of physical models and its capability to perform both 2D and 3D simulations [23]. In this study, Sentaurus TCAD was selected for its versatility in handling complex physical models. We utilized four main tools within Sentaurus TCAD: Sentaurus Structure Editor, Sentaurus Device, Sentaurus Visual, and Sentaurus Workbench. Sentaurus Visual was particularly instrumental for visualizing results and importing data, while the Sentaurus Workbench facilitated integration of various simulation components. The geometric model of the solar cell was constructed using Sentaurus Structure Editor. Figure 2 illustrates the geometric structure of the solar cell, highlighting variables such as the thickness of the electron transport layer (ETL), hole transport layer (HTL), and perovskite layers, which were adjusted within a specified range. Additionally, the doping concentrations and types of these layers (HTL, ETL, and perovskite) were defined within Sentaurus Structure Editor, set respectively at 1 × 10^16^ cm^−3^, 1 × 10^17^ cm^−3^, and 1 × 10^18^ cm^−3^. In numerical simulations, the mesh size significantly influences the result reliability. Our structure was meshed using two different sizes: a smaller mesh size of 1 nm for active regions such as heterojunctions, and a larger mesh size of 2 nm for other regions.

After creating the geometric model, the next step is to define the physical properties of each material used in the solar cell through numerical simulation in the Sentaurus Device. The Sentaurus Device allows us to control physical models, simulation accuracy, light intensity, temperature, and contacts. Choosing the right physical models from the options in Sentaurus TCAD is crucial, as using inappropriate ones can lead to unexpected errors in the simulation process. Simulating solar cells typically involves two stages: optical and electrical simulation. In optical simulation, we calculate important factors like photogenerated electron–hole pairs, how much light is absorbed, and how much is reflected or transmitted. There are three main optical models: the Transfer Matrix Method (TMM), Ray Tracing, and Beam Propagation. Each model has its advantages; for example, TMM is good for including interference effects in thin layers [24], while Ray Tracing is useful for textured solar cells because it tracks how light rays move through them [25]. In our study of planar solar cells, where we’re focusing on varying the thickness of each layer, we chose to use TMM (as defined by Formula (1)) for optical simulation. This model was chosen because it accurately includes interference effects, which is important for understanding how changing layer thickness affects solar cell performance.
(1)$$\left[\begin{array}{l} E_{i}\\ E_{r} \end{array}\right]=M\left[\begin{array}{l} E_{t}\\ 0 \end{array}\right]$$

Here, *M* is the matrix, *E_i_* is the electrical field of incident light, *E_r_* is the electric field of reflected light, and *E_t_* is the electric field of transmitted light.

In solar cells, ions, electrons, and holes play key roles in generating an electric field. The electrical field and potential within the solar cell are determined by solving the Poisson equation, represented by Formula (2). This equation takes into account the distribution of electrons and holes using the Fermi distribution to calculate their concentrations.
(2)$$\Delta \varphi =-\frac{q}{\varepsilon }\left(p-n+N_{D}+N_{A}\right)$$

Here, *ε* is the permittivity, *N_D_* and *N_A_* are the concentrations of donor and acceptor, q is the charge.

Electrons and holes in solar cells are separated by the internal electric field created by ions, after which they begin to migrate towards the electrodes. Various models exist to describe the transport of these charge carriers, such as drift–diffusion, thermodynamic, and hydrodynamic models. The thermodynamic and hydrodynamic models account for the influence of temperature on charge carrier transport. However, in our study, we did not specifically investigate the temperature effects on solar cell performance. Therefore, we employed the drift–diffusion model [26], as expressed by Formula (3), to calculate charge carrier transport.
(3)$$\begin{array}{l} J_{n}=-nq\mu_{n}\nabla \Phi_{n}\\ J_{p}=-pq\mu_{p}\nabla \Phi_{p} \end{array}$$

Here, *µ_n_*, *µ_p_* are the mobilities of electron and holes, *F_n_*, *F_p_* are the electron and hole quasi-Fermi potentials, *P_n_*, *P_p_* are the thermoelectric power of electrons and holes, *T* is the absolute temperature.

Photogenerated electrons and holes in solar cells can recombine through various mechanisms, such as defects within the material or by emitting photons. There are three main types of recombination: radiative recombination, Shockley–Read–Hall recombination, and Auger recombination. These mechanisms are all considered in our simulation to accurately model the behavior of charge carriers. Additionally, as an electrical boundary condition, the ohmic boundary condition [27] described by Formula (4) is utilized. This condition helps define how the electrical potential behaves at the boundaries of the solar cell structure, ensuring consistency and accuracy in our simulations.
(4)$$\begin{array}{l} \varphi =\varphi_{F}+\frac{kT}{q}a\sinh\left(\frac{N_{D}-N_{A}}{2n_{i,eff}}\right)\\ n_{0}p_{0}=n_{i,eff}^{2}\\ n_{0}=\sqrt{\frac{{\left(N_{D}-N_{A}\right)}^{2}}{4}+n_{i,eff}^{2}}+\frac{N_{D}-N_{A}}{2}\\ p_{0}=\sqrt{\frac{{\left(N_{D}-N_{A}\right)}^{2}}{4}+n_{i,eff}^{2}}-\frac{N_{D}-N_{A}}{2} \end{array}$$

Here, *n_i,eff_* is concentration of effective intrinsic carrier, *φ_F_* is Fermi potential of contact.

## 3. Results and Discussion

### 3.1. Optimization of ETL

We studied the NiO_x_/perovskite/MeO structure (MeO = TiO_2_, ZnO, MoO_3_) to optimize geometrical sizes. The thicknesses of the NiO_x_ and perovskite layers were set at 100 nm and 200 nm, respectively, while the ETL layer thickness varied from 20 nm to 200 nm. We calculated photoelectric parameters for each thickness. Figure 3 shows the dependence of a short circuit current (a) and output power (b) on the ETL layer thickness. Both parameters exhibited a peak value within the 20–200 nm range. When comparing ETL layers, ZnO had the highest short circuit current, while TiO_2_ had the lowest. In an experiment, it was found that perovskite solar cell with ZnO as ETL has a higher short circuit current [28]. However, in terms of output power, TiO_2_ demonstrated the highest value, and MoO_3_ had the lowest. Notably, the output power of solar cells with ZnO and TiO_2_ was nearly the same, but it was significantly lower in case of MoO_3_. This discrepancy between output power and short circuit current is unusual, as a high short circuit current typically correlates with better output power. The reason for TiO_2_’s superior output power, despite its lower short circuit current, may be attributed to its higher fill factor and efficient charge extraction properties, which offset the lower current generated.

To further investigate this inconsistency, we examined the dependence of the fill factor on ETL layer thickness, as shown in Figure 4. The fill factor did not change significantly with thickness variations: the amplitude of change for MoO_3_ was less than 0.1%; for TiO_2_, less than 0.01%; and for ZnO, less than 0.08%. Among all ETL layers, the solar cell with TiO_2_ exhibited the highest fill factor, while the one with MoO_3_ had the lowest. This high fill factor in the TiO_2_ layer is crucial, as it allows the solar cell to achieve maximum output power despite having the lowest short circuit current. The stability of the fill factor across different thicknesses further emphasizes the robustness of TiO_2_ as an effective ETL material in this structure. The high fill factor of TiO_2_ can be attributed to its excellent electronic properties, such as high electron mobility and favorable energy level alignment with the perovskite layer, which reduces recombination losses and improves charge collection efficiency. In an experiment, Wang also found [29] that TiO_2_ ETL enhances the extraction and transportation of electrons to contacts from perovskite. So, our simulation results agree with the experiment. On the other hand, the low fill factor of MoO_3_ might be due to its lower electron mobility and potential issues with charge recombination at the interface, leading to less efficient charge extraction.

Based on the analysis, the optimal thicknesses of the ETL layers for maximum output power were identified: 80 nm for MoO_3_, 60 nm for TiO_2_, and 100 nm for ZnO. Table 1 provides the photoelectric parameters of solar cells with various ETL layers at these optimal thicknesses. The short circuit current for ZnO, MoO_3_, and TiO_2_ layers at optimal thicknesses was 15.79 mA/cm^2^, 15.62 mA/cm^2^, and 15.52 mA/cm^2^, respectively. The open circuit voltage was approximately 1.2 V for all ETLs. The fill factor was 54.18% for MoO_3_, 84.47% for TiO_2_, and 82.15% for ZnO. Consistent with our previous studies, TiO_2_ demonstrated a superior fill factor compared to other ETL materials. The efficiency of solar cells with optimal ETL layer thicknesses was 16.04% for MoO_3_, 25.61% for TiO_2_, and 24.6% for ZnO. These results confirm that TiO_2_ is the best ETL layer for the inverted perovskite structure, combining a high fill factor and efficiency, making it an excellent choice for optimizing solar cell performance.

### 3.2. Optimization of HTL

After finding the optimal thickness of ETL layers for each material, we decided to determine the optimal thickness of HTL layers. In this case, we maintained the optimal thickness of ETL layers and varied the NiO_x_ layer thickness from 20 nm to 200 nm. Figure 5 presents the dependence of a short circuit current (a) and output power (b) on NiO_x_ layer thickness. Again, the short circuit current of the solar cell with ZnO as the ETL layer is the highest, and the lowest belongs to TiO_2_ as the ETL layer. However, the output power of the solar cell with TiO_2_ as the ETL layer is the highest, while the lowest belongs to MoO_3_. Both short circuit current and output power functions depending on NiO_x_ layer thickness have an extremum in the range of 20–200 nm, indicating that the optimal value of the HTL layer thickness lies within this range. The increase and subsequent decrease in performance with varying thickness are due to the balance between photogeneration and recombination. Increasing the ETL or HTL layer thickness helps capture more photons but also leads to increased recombination. Therefore, ETL and HTL layers should be thick enough to transmit electrons or holes with a lower recombination rate to the contacts. According to the output power, the optimum value of the HTL layers is 80 nm for all three solar cells, suggesting that the type of ETL layer does not affect the optimal value of the HTL layer, likely because the sun rays fall from the HTL layer side. In experiment [30], it was predicted that optimal thickness of NiO_x_ is between 50–150 nm. Experimental optimal thickness range includes our optimal thickness obtained by simulation and it shows agreement of our results with the experiment.

The thickness of the HTL layer can affect the resistivity of the solar cell. As the ETL or HTL layer thickness increases, so does the resistivity, which in turn affects the fill factor of the solar cell. Figure 6 shows the dependence of the fill factor on the thickness of the HTL layer. Unlike the ETL layer thickness, the fill factor changed significantly with changes in HTL layer thickness. The amplitude of fill factor fluctuation was 0.7% for MoO_3_, 1.4% for TiO_2_, and 0.7% for ZnO. The solar cell with TiO_2_ had the highest fill factor, while the lowest belonged to MoO_3_. This explains why the output power of the TiO_2_-based solar cell is the highest, even though its short circuit current is the lowest. The fill factor significantly determines the optimal material for the ETL layer in our study. The functional dependence of the fill factor of the solar cell with TiO_2_ as the ETL layer on HTL layer thickness had an extremum in the range of 20–200 nm, consistent with the short circuit current and output power. The quality of the curve for the functional dependence of the fill factor on HTL layer thickness for solar cells with ZnO and MoO_3_ was similar but with a lower fluctuation amplitude, indicating that they did not significantly affect the optimal value of HTL layer thickness where output power reaches a maximum value.

We provided the photoelectric parameters of solar cells with optimal ETL and HTL layer thicknesses in Table 2. The optimal thicknesses of ETL layers with MoO_3_, TiO_2_, ZnO, and HTL with NiO_x_ are 80 nm, 60 nm, 100 nm, and 80 nm, respectively. The short circuit current of solar cells with optimal ETL and HTL layer thicknesses is 15.92 mA/cm^2^ for MoO_3_, 15.84 mA/cm^2^ for TiO_2_, and 16.06 mA/cm^2^ for ZnO. The optimization of the HTL layer did not affect the open circuit voltage. The efficiency of solar cells with MoO_3_, TiO_2_, and ZnO is 16.35%, 26.15%, and 25.03%, respectively, with fill factors of 54.17%, 87.5%, and 82.21%, respectively. Both the fill factor and efficiency of the solar cell with MoO_3_ are 1.6 times lower than those of the solar cell with TiO_2_. The resistivity of TiO_2_ [31], MoO_3_ [32], and ZnO [33] varies with thickness, with average values of 1 Ωcm, 5.4 Ωcm, and 2 Ω·cm, respectively. Because of the high resistivity of MoO_3_, its efficiency and fill factor is the lowest.

### 3.3. Optimization of Perovskite Layer

After optimizing the HTL layer thickness, we observed efficiency increases of 0.31%, 0.54%, and 0.43% for MoO_3_, TiO_2_, and ZnO, respectively. Despite these improvements, the efficiency gains were lower than expected by optimizing only the ETL and HTL layers. Therefore, we decided to optimize the perovskite layer as well. We varied the thickness of the perovskite layer from 200 nm to 1600 nm while keeping the optimal ETL and HTL layer thicknesses from previous calculations. Figure 7 shows the dependence of a short circuit current (a) and output power (b) on perovskite layer thickness. The short circuit current increased with the perovskite layer thickness and saturated after 1200 nm. This saturation occurs due to the balance between photogeneration and recombination rates. As the perovskite layer thickness increases, both absorbed photon density and recombination increase. Initially, the short circuit current of the solar cell with TiO_2_ was the lowest, but after increasing the thickness, it became the highest. The short circuit current increased from 15.8 mA/cm^2^ to 22.6 mA/cm^2^ when the perovskite layer thickness was increased from 200 nm to 1600 nm. The output power of each solar cell showed a maximum at a specific perovskite layer thickness. For the solar cell with MoO_3_, the output power increased from 10.4 mW/cm^2^ to 11.8 mW/cm^2^ when the perovskite layer thickness increased from 200 nm to 600 nm, after which it began to decrease. The output power of the solar cell with ZnO also increased from 15.8 mW/cm^2^ to 19.1 mW/cm^2^ within the perovskite layer thickness range of 200–600 nm. The output power of the solar cell with TiO_2_ behaved differently; it increased from 16.5 mW/cm^2^ to 22.3 mW/cm^2^ in the range of 200–800 nm and then began to saturate, showing little change beyond 800 nm. Based on these observations, we identified the optimal perovskite layer thickness for solar cells with MoO_3_, ZnO, and TiO_2_ as 600 nm, 600 nm, and 800 nm, respectively. For the solar cells with MoO_3_ and ZnO, the optimal perovskite thickness corresponded to the point of maximum output power. For the solar cell with TiO_2_, the optimal thickness was determined at the point where the output power started to saturate.

In Figure 8, the dependence of the fill factor on perovskite layer thickness is shown. As the thickness of the perovskite layer increases from 200 nm to 1600 nm, the fill factor of solar cells with MoO_3_, TiO_2_, and ZnO decreased by 14.4%, 1.9%, and 13.8%, respectively. Increasing the thickness of the perovskite layer leads to more photon absorption but also increases the recombination rate. The optimal thickness is determined by balancing recombination and photogeneration. The fill factor of solar cells with ZnO and MoO_3_ decreased significantly, which is the main reason for having an exact optimal thickness of the perovskite layer where the maximum output power is reached. On the other hand, the fill factor of the solar cell with TiO_2_ decreased only slightly, leading to the saturation of output power rather than a distinct maximum as the perovskite layer thickness increased. This suggests that TiO_2_ is the best ETL for inverted perovskite solar cells. The slight decrease in the fill factor for the solar cell with TiO_2_ indicates better charge transport and lower fill factor compared to ZnO and MoO_3_.

Each layer of the inverted perovskite solar cell has been fully optimized. In Table 3, the photoelectric parameters of the solar cells with optimal ETL, HTL, and perovskite layers are provided. There is no significant difference in the short circuit currents among the solar cells. The short circuit currents for solar cells with MoO_3_, TiO_2_, and ZnO are 21.12 mA/cm^2^, 21.83 mA/cm^2^, and 21.2 mA/cm^2^, respectively. Before optimizing the perovskite layer thickness, the short circuit current of the solar cell with TiO_2_ was the lowest; after optimization, it became the highest. Due to the increased thickness of the perovskite layer, the open circuit voltage decreased by 0.012 V. The fill factors for solar cells with MoO_3_, TiO_2_, and ZnO are 44.14%, 86.83%, and 75.78%, respectively. The fill factor of the solar cell with TiO2 is almost twice that of the solar cell with MoO_3_. Consequently, the efficiencies of solar cells with MoO_3_, TiO_2_, and ZnO are 18.69%, 35.21%, and 30.16%, respectively. Similar to the fill factor, the efficiency of the solar cell with TiO_2_ is twice that of the solar cell with MoO_3_. Azri’s simulation study [34] also revealed that ZnO and TiO_2_ are the best candidates for ETL. ETL layers primarily transport electrons, and the band offset of conduction bands between the perovskite and ETL plays a vital role. The band offsets for MoO_3_, TiO_2_, and ZnO with the perovskite are 2.7 eV, 0.3 eV, and 0.6 eV, respectively. The large band offset for MoO_3_ could be the main reason for the low fill factor of the solar cell with MoO_3_. The optimization process highlights the critical importance of selecting and tuning the ETL, HTL, and perovskite layers to maximize the performance of inverted perovskite solar cells. Despite the initial lower short circuit current, the solar cell with TiO_2_ demonstrated the highest overall efficiency and fill factor, underscoring its superiority as an ETL material. The significant differences in band offsets among the materials suggest that minimizing the band offset between the perovskite and ETL is crucial for achieving higher fill factors and efficiencies.

### 3.4. Comparision of Traditional and Inverted Perovskite Solar Cells

Our main mission was to determine the better structure for perovskite solar cells, whether inverted or traditional. In Table 4, we present the photoelectric parameters of traditional solar cells with the same thicknesses of ETL, HTL, and perovskite layers as those in the inverted structure. We compared the photoelectric parameters of traditional and inverted perovskite solar cells. The short circuit currents of traditional solar cells with MoO_3_, TiO_2_, and ZnO are 1.9 mA/cm^2^, 3.5 mA/cm^2^, and 1.2 mA/cm^2^ smaller than those of inverted solar cells. This indicates that the solar cell with ZnO has the highest short circuit current among the traditional configurations. This difference can be explained by the optical properties of the materials used. While the open circuit voltage of traditional solar cells is also smaller, the fill factor is greater compared to inverted solar cells. The efficiencies of traditional solar cells with MoO_3_, TiO_2_, and ZnO are 1.5%, 5.7%, and 1.5% lower than those of the inverted solar cells. The NiO_x_-based inverted perovskite solar cell achieved more than 20% efficiency in the experiment and it was higher than that of the traditional one. These results suggest that the inverted structure generally provides better performance in terms of short circuit current and efficiency, despite the traditional structure showing a higher fill factor. This comprehensive comparison highlights the advantages of the inverted configuration in maximizing the overall efficiency and photoelectric performance of perovskite solar cells.

When we change the illumination side, it significantly affects the optical properties of the solar cells. The differences in efficiency and short circuit current between inverted and traditional perovskite solar cells can be attributed to these optical properties. In Figure 9, we present the absorption and reflection spectra of both inverted and traditional perovskite solar cells. In the wavelength range of 0.5–0.8 µm, the absorption coefficient of the inverted solar cell with TiO_2_ is higher than that of the other solar cells. This higher absorption is due to a lower reflection coefficient in this range. The improved absorption and reflection properties of the inverted solar cell with TiO_2_ contribute to its higher efficiency. The enhanced optical properties, combined with better electronic properties, explain why the inverted structure with TiO_2_ performs better in terms of efficiency and short circuit current.

In the case of traditional perovskite solar cells, the absorption coefficient of the solar cell with TiO_2_ is smaller than that of other solar cells. Additionally, its reflection coefficient is higher. In traditional solar cells, the optical properties of ETL layers play a crucial role in determining the absorption and reflection coefficients. In Figure 10, we present the real (a) and imaginary (b) parts of the complex refractive indices of ZnO, TiO_2_, MoO_3_, NiO_x_, and perovskite. The average refractive indices of ZnO, TiO_2_, MoO_3_, NiO_x_, and perovskite are 1.9, 1.92, 2.2, 1.65, and 2.62, respectively. Given that the refractive index of air is 1, the theoretical optimal refractive index for a layer between air and perovskite should be 1.61. Therefore, based on refractive indices, NiO_x_ is the best candidate. In an inverted perovskite solar cell, light falls on solar cells from the NiO_x_ side. In this configuration, NiO_x_ acts as an effective antireflection coating in addition to its role in hole transport. Moreover, in an experiment, Wu found [35] that a good device lifetime of inverted perovskite solar cells with NiO_x_ as HTL can be stable for up to 50 days.

The refractive indices of ETL materials are higher than the optimal refractive index for antireflection purposes. As a result, the absorption coefficient of the traditional solar cell is smaller than that of the inverted solar cell. The maximum absorption coefficients of inverted and traditional perovskite solar cells in the wavelength range of 0.3–0.9 µm are 1 and 0.85, respectively. Therefore, the efficiency of the inverted solar cell is higher than that of the traditional solar cell due to the higher absorption coefficient.

## 4. Conclusions

In conclusion, our study investigated the performance of inverted perovskite solar cells by optimizing the thickness of their layers. We focused on three materials for the electron transport layer (ETL)—ZnO, MoO_3_, and TiO_2_—and found that TiO_2_ provided the best results. The performance was evaluated based on a short circuit current, output power, and fill factor. TiO_2_ showed the highest efficiency and fill factor, making it the best choice for the ETL. The study also examined the hole transport layer (HTL) and the perovskite layer. We found that the optimal thickness for the HTL was 80 nm for all tested ETL materials, while the optimal thickness for the perovskite layer varied: 600 nm for MoO_3_ and ZnO, and 800 nm for TiO_2_. After optimizing these layers, the solar cell with TiO_2_ as the ETL achieved the highest efficiency of 35.21%, significantly outperforming the cells with ZnO and MoO_3_. This shows the importance of selecting and fine-tuning the materials and thicknesses of each layer to maximize the performance of perovskite solar cells. Our findings highlight that TiO_2_ is the most effective ETL material due to its excellent electronic properties and alignment with the perovskite layer, which enhances charge collection and reduces recombination losses. This optimization is crucial for improving the efficiency and stability of perovskite solar cells, making them more viable for large-scale energy production.

## Figures and Tables

**Figure 1 nanomaterials-14-01301-f001:**
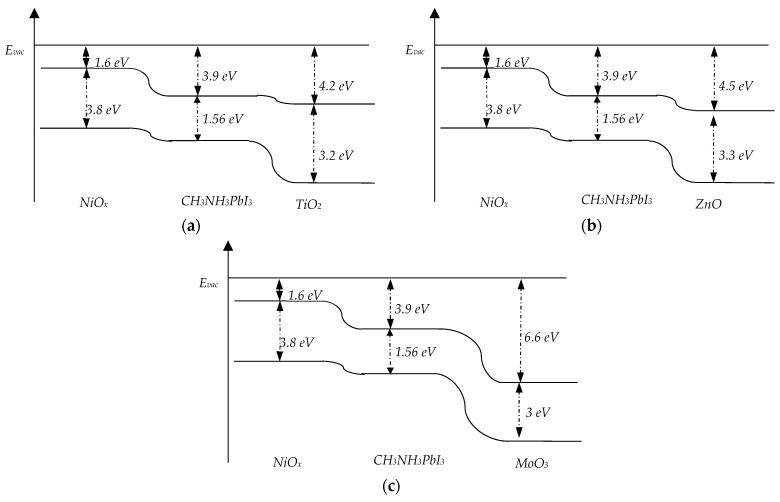
Band structure of NiO_x_/CH_3_NH_3_PbI_3_/TiO_2_ (**a**), NiO_x_/CH_3_NH_3_PbI_3_/ZnO (**b**), and NiO_x_/CH_3_NH_3_PbI_3_/MoO_3_ (**c**) solar cells.

**Figure 2 nanomaterials-14-01301-f002:**
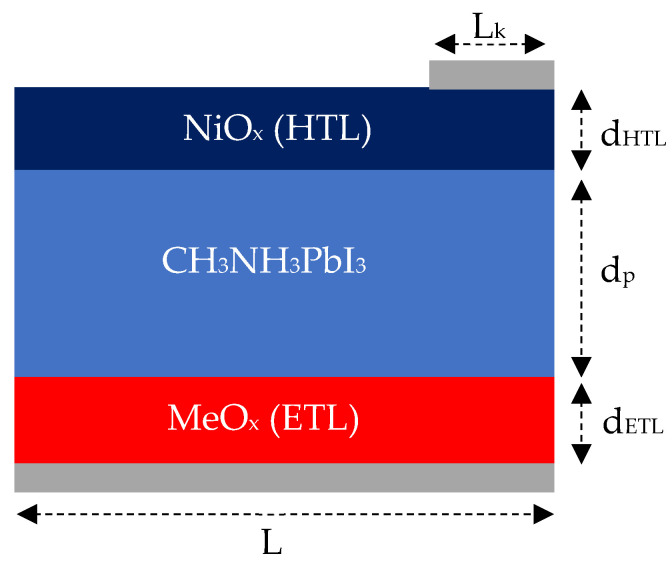
Geometrical structure of inverted perovskite solar cell. *L_k_*—width of contact region, *d_HTL_*—thickness of HTL, *d_p_*—Thickness of perovskite layer, *d_ETL_*—thickness of ETL, *L*—width of solar cell.

**Figure 3 nanomaterials-14-01301-f003:**
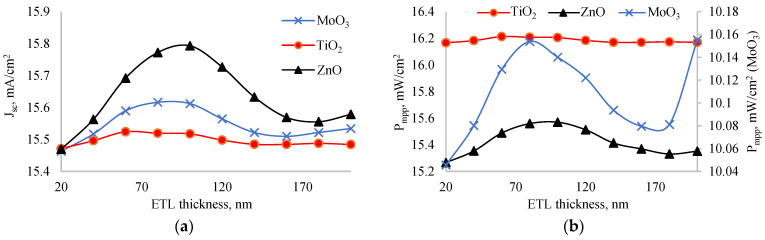
Dependence of short circuit current (**a**) and output power (**b**) on thickness of ETL.

**Figure 4 nanomaterials-14-01301-f004:**
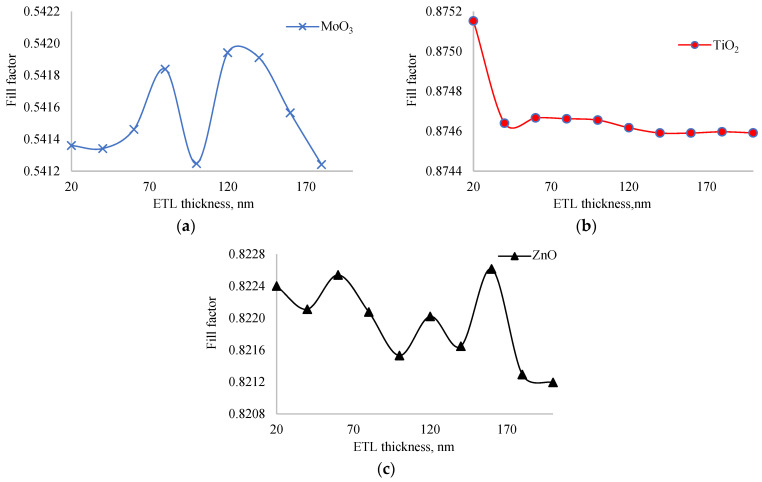
Dependence of fill factor of solar cells with MoO_3_ (**a**), TiO_2_ (**b**) and ZnO (**c**) as ETL on thickness of ETL.

**Figure 5 nanomaterials-14-01301-f005:**
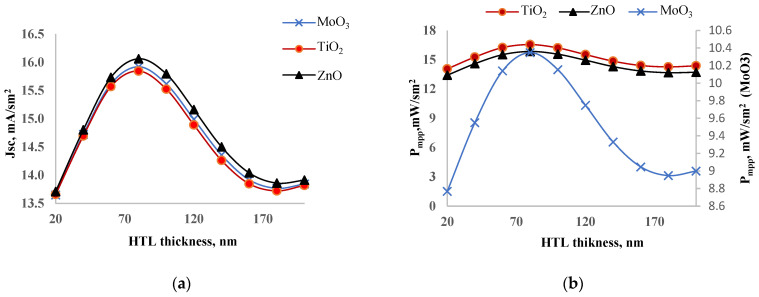
Dependence of short circuit current (**a**) and output power (**b**) on thickness of HTL.

**Figure 6 nanomaterials-14-01301-f006:**
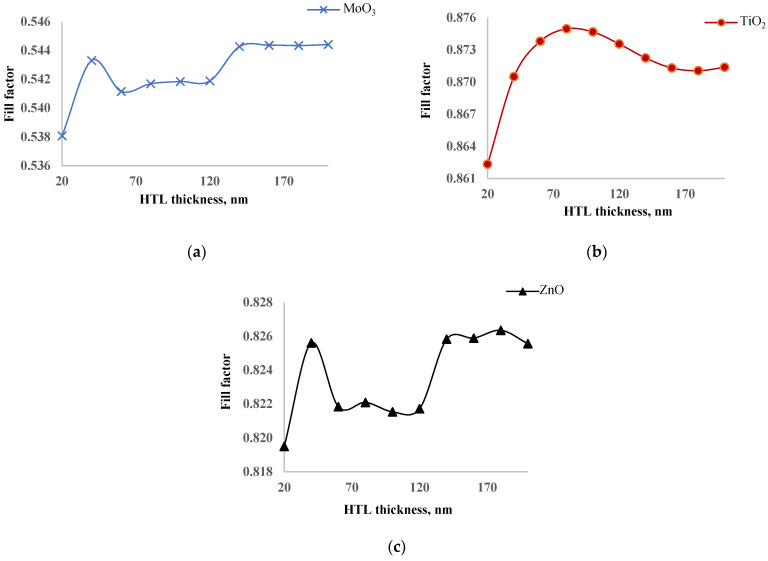
Dependence of fill factor of solar cells with MoO_3_ (**a**), TiO_2_ (**b**), and ZnO (**c**) as ETL on thickness of HTL.

**Figure 7 nanomaterials-14-01301-f007:**
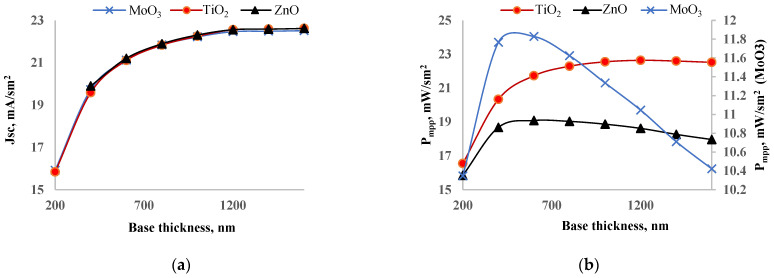
Dependence of short circuit current (**a**) and output power (**b**) on thickness of perovskite layer thickness.

**Figure 8 nanomaterials-14-01301-f008:**
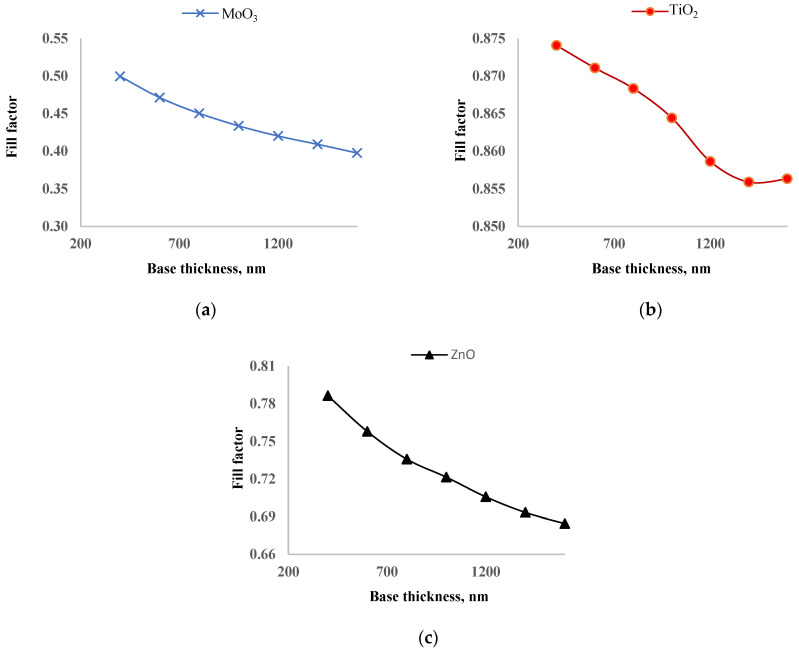
Dependence of fill factor of solar cells with MoO_3_ (**a**), TiO_2_ (**b**), and ZnO (**c**) as ETL on thickness of perovskite layer.

**Figure 9 nanomaterials-14-01301-f009:**
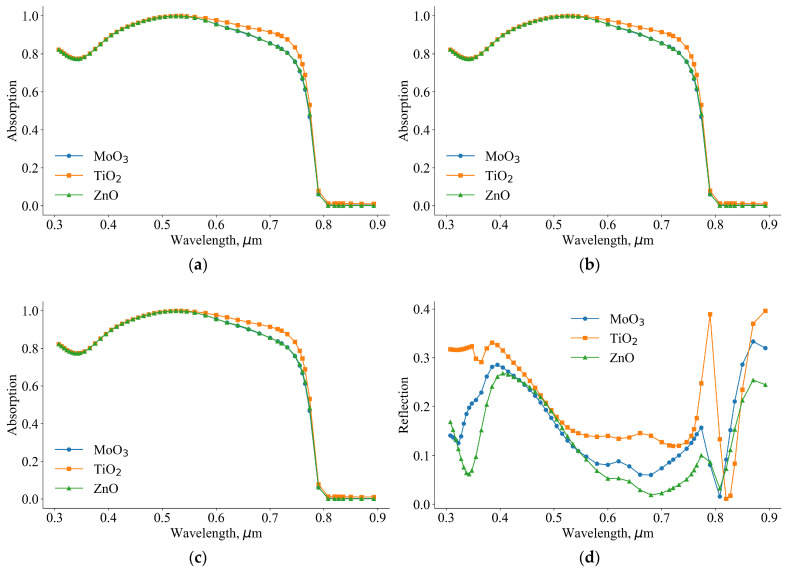
Absorption (**a**,**c**) and reflection (**b**,**d**) spectrum of inverted (**a**,**b**) and traditional (**c**,**d**) perovskite solar cells.

**Figure 10 nanomaterials-14-01301-f010:**
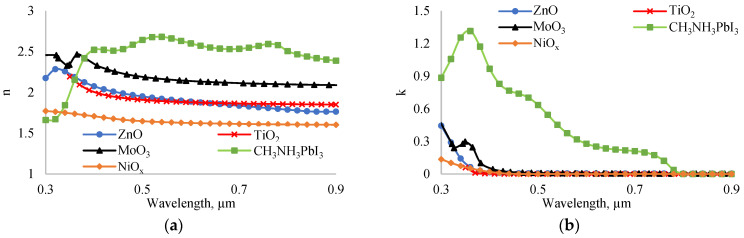
Dependence of real (**a**) and imaginary (**b**) part of complex refractive indices of ZnO, MoO_3_, NiO_x_, TiO_2_ and CH_3_NH_3_PbI_3_ on wavelength.

**Table 1 nanomaterials-14-01301-t001:** Photoelectric parameters of inverted perovskite solar cell with optimal ETL thickness. *d_base_* = 200 nm and *d_HTL_* = 100 nm.

Materials	*d_ETL_*, nm	J_sc_, mA/cm^2^	U_oc_, V	P_mpp_, mW/cm^2^	FF, %	η, %
MoO_3_	80	15.62	1.200	10.15	54.18	16.04
TiO_2_	60	15.52	1.194	16.21	87.47	25.61
ZnO	100	15.79	1.200	15.57	82.15	24.60

**Table 2 nanomaterials-14-01301-t002:** Photoelectric parameters of inverted perovskite solar cell with optimal ETL and HTL thicknesses. *d_base_* = 200 nm.

Materials	*d_ETL_*, nm	*d_HTL_*, nm	J_SC_, mA/sm^2^	U_OC_, V	P_mpp_, mW/sm^2^	FF, %	η, %
MoO_3_	80	80	15.92	1.200	10.35	54.17	16.35
TiO_2_	60	80	15.84	1.194	16.55	87.50	26.15
ZnO	100	80	16.06	1.200	15.84	82.21	25.03

**Table 3 nanomaterials-14-01301-t003:** Photoelectric parameters of inverted perovskite solar cell with optimal ETL, HTL, and perovskite layer thicknesses.

Materials	d_BASE_, nm	*d_ETL_*, nm	*d_HTL_*, nm	J_SC_, mA/sm^2^	U_OC_, V	P_mpp_, mW/sm^2^	FF, %	η, %
MoO_3_	600	80	80	21.12	1.188	11.83	47.14	18.69
TiO_2_	800	60	80	21.83	1.176	22.29	86.83	35.21
ZnO	600	100	80	21.20	1.188	19.09	75.78	30.16

**Table 4 nanomaterials-14-01301-t004:** Photoelectric parameters of traditional perovskite solar cell with optimal ETL and HTL thicknesses.

Materials	d_BASE_, nm	*d_ETL_*, nm	*d_HTL_*, nm	J_SC_, mA/sm^2^	U_OC_, V	P_mpp_, mW/sm^2^	FF, %	η, %
MoO_3_	600	80	80	19.20	1.182	10.86	47.85	17.15
TiO_2_	800	60	80	18.30	1.171	18.68	87.14	29.51
ZnO	600	100	80	20.01	1.182	18.16	76.81	28.69

## Data Availability

Data will be made available upon request from the corresponding author.

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
