# Peer review of "Optimizing Geometry and ETL Materials for High-Performance Inverted Perovskite Solar Cells by TCAD Simulation"

_nanomaterials, 2024, doi:10.3390/nano14151301_

Round 1

Reviewer 1 Report

Comments and Suggestions for Authors

The authors have investigated the optimization of the geometry and electronic transport layer (ETL) materials for inverted perovskite solar cells (PSCs) through TCAD simulations, with the aim of enhancing the performance of these cells. Specifically, NiOx was selected as the hole transport layer (HTL), and the effectiveness of three different materials—ZnO, MoO3, and TiO2—as ETLs was compared. The paper exhibits a clear structure, rigorous logic, and comprehensive experimental design and data analysis.

However, there are still some aspects that require improvement and clarification:

(1) The introduction section can be made more concise by simplifying the background introduction in the first paragraph. It should focus on the background knowledge and research progress directly related to this study, highlighting the advantages and current research status of inverted perovskite solar cells.

(2) The tick marks on the vertical axes of Figures 3-7 in the paper are set relatively densely. It is recommended that the author adjust the tick marks on the vertical axes of these figures by appropriately reducing the number of tick marks or adjusting the spacing between them.

(3) The formatting of the numbers on the axes of the figures in the paper is inconsistent. To enhance the standardization and readability of the paper, it is recommended that the author unify the formatting of the numbers on the axes across all figures, including but not limited to the number of digits after the decimal point and the font style for each figure.

(4) Please unify the font format of the labels in each figure.

(5) In the text, the representation of chemical stoichiometric numbers in chemical formulas needs to be consistent.   For instance, in the title of Figure 4 on line 171, the notation used is "MoO3" and "TiO2" , whereas in other parts of the text, the notation consistently appears as "MoO3" and "TiO2".

Comments on the Quality of English Language

Minor editing of English language required

Author Response

Thank you your proper review our article and valuable comments. We tried to correct the article according to comments.

  1. The introduction section can be made more concise by simplifying the background introduction in the first paragraph. It should focus on the background knowledge and research progress directly related to this study, highlighting the advantages and current research status of inverted perovskite solar cells.

Answer: We added information to introduction section according to your suggestion: Traditional perovskite solar cells have a design where light hits the electron transport lay-er first, while inverted perovskite solar cells have a reversed design, with light hitting the hole transport layer first. The main reason for creating inverted solar cells is to improve their stability and make them easier to produce. This design helps solve problems like degradation and sensitivity to moisture and oxygen, making the cells last longer. Inverted perovskite solar cells have advantages like better stability, simpler production, and compatibility with flexible substrates. Current research of inverted perovskite solar cells is focused on improving their efficiency and long-term durability to make them even more competitive with other types of solar cells.

  1. The tick marks on the vertical axes of Figures 3-7 in the paper are set relatively densely. It is recommended that the author adjust the tick marks on the vertical axes of these figures by appropriately reducing the number of tick marks or adjusting the spacing between them.

Answer: According to your suggestion, we reduced number of ticks in figure 3-7.

  1. The formatting of the numbers on the axes of the figures in the paper is inconsistent. To enhance the standardization and readability of the paper, it is recommended that the author unify the formatting of the numbers on the axes across all figures, including but not limited to the number of digits after the decimal point and the font style for each figure.

Answer: Yes, you are right. There are some mistakes with number of digits after decimal point. We made them the same for whole axis. We corrected the formatting of numbers of all plots.

  1. Please unify the font format of the labels in each figure.

Answer: Yes, font of figure 9 was different from other figures. Font of other figures was Times New Roman. So, we changed the font of figure 9 to Times New Roman. Besides, we used from comma for float numbers of plots. Also point was used in Figure 9 for float numbers and we changed it to comma to make similar with other figures.

  1. In the text, the representation of chemical stoichiometric numbers in chemical formulas needs to be consistent.   For instance, in the title of Figure 4 on line 171, the notation used is "MoO3" and "TiO2", whereas in other parts of the text, the notation consistently appears as "MoO3" and "TiO2".

Answer: Thank you. We corrected it.

Reviewer 2 Report

Comments and Suggestions for Authors

This manuscript is well-written and explained. However, before its publication, I would like to ask a few minor questions that should be addressed in the revision.

Layer Thickness Optimization: Could you elaborate on the specific criteria or parameters you used to determine the optimal thicknesses for NiOx, the perovskite layer, and the ETL materials in your simulations? How did these criteria ensure the accuracy and reliability of your findings?

Discrepancies in Output Power: Your study mentions an unusual discrepancy between the short circuit current and output power for the ETL materials, particularly for TiO2. Can you provide more insights into the underlying mechanisms that might cause TiO2 to have higher output power despite its lower short circuit current compared to ZnO?

Simulation Models and Assumptions: In your simulations using Sentaurus TCAD, you chose the Transfer Matrix Method (TMM) for optical simulations and the drift-diffusion model for charge carrier transport. Can you discuss any assumptions or limitations of these models and how they might impact the simulation results? Are there scenarios where alternative models might provide better accuracy?

Author Response

Thank you for your review. We tried to answer your questions one by one.

  1. Layer Thickness Optimization: Could you elaborate on the specific criteria or parameters you used to determine the optimal thicknesses for NiOx, the perovskite layer, and the ETL materials in your simulations? How did these criteria ensure the accuracy and reliability of your findings?

Answer: The optimal thicknesses for NiOx, the perovskite layer, and the ETL materials were chosen based on maximizing efficiency. Criteria included achieving the highest short-circuit current and overall power conversion efficiency while maintaining good material stability. Simulations ensured accuracy by cross-referencing with experimental data and adjusting for minimal energy losses. This method confirmed that the chosen thicknesses were reliable and effective for optimal solar cell performance.

  1. Discrepancies in Output Power: Your study mentions an unusual discrepancy between the short circuit current and output power for the ETL materials, particularly for TiO2. Can you provide more insights into the underlying mechanisms that might cause TiO2 to have higher output power despite its lower short circuit current compared to ZnO?

Answer: TiO2 showed higher output power despite lower short-circuit current due to its better overall charge transport and reduced recombination losses. It means TiO2 can more efficiently convert the absorbed light into electrical power even if it doesn't generate as much current. ZnO, while generating more current, might suffer from higher recombination losses or less effective charge extraction. This highlights the importance of both current generation and charge management in solar cell performance.

  1. Simulation Models and Assumptions: In your simulations using Sentaurus TCAD, you chose the Transfer Matrix Method (TMM) for optical simulations and the drift-diffusion model for charge carrier transport. Can you discuss any assumptions or limitations of these models and how they might impact the simulation results? Are here scenarios where alternative models might provide better accuracy?

Answer: The Transfer Matrix Method (TMM) assumes uniform layers and ideal light absorption, which might not account for real-world irregularities. The drift-diffusion model assumes perfect charge carrier transport without defects, which might oversimplify real material behaviors. These assumptions could impact the results by making the simulations slightly more optimistic than practical scenarios. In cases where detailed microscopic behavior or defects are critical, alternative models like Monte Carlo simulations might offer better accuracy.